# Dose-Dependent Effect of DNA Vaccine pVAX-H5 Encoding a Modified Hemagglutinin of Influenza A (H5N8) and Its Cross-Reactivity Against A (H5N1) Influenza Viruses of Clade 2.3.4.4b

**DOI:** 10.3390/v17030330

**Published:** 2025-02-27

**Authors:** Andrey P. Rudometov, Victoria R. Litvinova, Andrei S. Gudymo, Ksenia I. Ivanova, Nadezhda B. Rudometova, Denis N. Kisakov, Mariya B. Borgoyakova, Lyubov A. Kisakova, Vladimir A. Yakovlev, Elena V. Tigeeva, Danil I. Vahitov, Kristina P. Makarova, Natalia P. Kolosova, Tatiana N. Ilyicheva, Vasiliy Yu. Marchenko, Artemiy A. Sergeev, Larisa I. Karpenko, Alexander A. Ilyichev

**Affiliations:** Federal Budgetary Research Institution State Research Center of Virology and Biotechnology Vector, Rospotrebnadzor, 630559 Koltsovo, Russia

**Keywords:** DNA vaccines, influenza A (H5N8) virus, influenza A (H5N1) virus, jet injection, immunogenicity, microneutralization, protectivity

## Abstract

Highly pathogenic avian influenza (HPAI) H5 clade 2.3.4.4b viruses are widespread in wild and domestic birds, causing severe economic damage to the global poultry industry. Moreover, viruses of this clade are known to cause infections in mammals, posing a potential pandemic threat. Due to the ongoing evolution and change in the dominant strains of H5 clade 2.3.4.4b, it is important to investigate the cross-reactivity of vaccines in use and under development against clade 2.3.4.4b viruses. In this study, the immunogenicity of the previously developed DNA vaccine encoding a modified hemagglutinin of the influenza A/turkey/Stavropol/320-01/2020 (H5N8) virus, administered by jet injection at doses of 1, 10, 50, 100, and 200 μg, was investigated. The highest titer of specific to recombinant hemagglutinin antibodies was detected in the group of animals injected with 100 µg of DNA vaccine. The cross-reactivity study of sera of animals immunized with 100 µg of DNA vaccine in a microneutralization assay against the strains A/chicken/Astrakhan/321-05/2020 (H5N8), A/chicken/Komi/24-4V/2023 (H5N1), and A/chicken/Khabarovsk/24-1V/2022 (H5N1) showed the formation of cross-neutralizing antibodies. Moreover, the study of protective properties showed that the DNA vaccine protected animals from mortality after infection with A/chicken/Khabarovsk/24-1V/2022 (H5N1) virus.

## 1. Introduction

Influenza A and B viruses cause seasonal epidemics, but only influenza A viruses have caused pandemics in the past with high mortality and economic costs [1,2]. The wide host range provides influenza A viruses with a high probability of genetic reassortment, resulting in zoonotic viruses that can spread to the human population. In particular, highly pathogenic avian influenza (HPAI) H5 viruses also have pandemic potential [3,4,5,6,7,8,9,10,11]. According to the WHO, HPAI H5 viruses have caused 904 human infections, including 464 deaths [12]. In 2020, a new H5N1 virus of clade 2.3.4.4b emerged as a result of reassortment, which by now, has become the most common variant of the highly pathogenic H5 influenza virus [13]. HPAIV H5N1 clade 2.3.4.4b has demonstrated an enhanced ability to cross species barriers and infect several mammalian species, including humans [14,15]. The host range of the virus has recently been expanded to include ruminants, particularly dairy cattle in the United States [16,17]. In addition, several cases of HPAIV H5N1 infection in humans have also been reported among dairy farm workers [18,19,20,21]. This is believed to be the first confirmed case of mammal-to-human transmission of HPAIV H5N1. In addition, it has been shown that a few amino acid substitutions (e.g., substitutions of Gln for Leu at position 226 in hemagglutinin [22]) are sufficient for A/H5 viruses to become airborne between mammals [23].

Thus, given the high incidence of infection and mortality in mammals [24,25,26,27,28], as well as the high variability of circulating HPAI H5 virus strains, it is necessary to conduct studies aimed at investigating the activity of existing and emerging vaccines against the different HPAI H5 strains of clade 2.3.4.4b [29]. In the study [30], it was demonstrated that licensed H5N1 vaccines derived from the HPAI H5N1 virus strains (A/Vietnam, clade 1 and A/Indonesia, clade 2.1) held in reserve for a pandemic in the United States were able to induce cross-neutralizing antibodies against the HPAI clade 2.3.4.4b A/Astrakhan/3212/2020 (H5N8) virus.

Previously, we developed DNA vaccine pVAX-H5 and investigated its immunogenicity against the A/turkey/Stavropol/320-01/2020 (H5N8) strain (homologous to the A/Astrakhan/3212/2020 (H5N8) strain), at a dose of 100 μg [28]. The A/Astrakhan/3212/2020 (H5N8) strain has been recommended by the WHO as a pandemic vaccine strain and has been available to vaccine manufacturers since 2022. This study presents the results of the dose-dependent effect of the DNA vaccine pVAX-H5 in the range of 1 to 200 μg and the cross-reactivity of the sera of animals immunized with 100 µg of DNA vaccine in a microneutralization assay against A/chicken/Komi/24-4V/2023 (H5N1) and A/chicken/Khabarovsk/24-1V/2022 (H5N1) viruses. The protective properties of the DNA vaccine against the A/chicken/Khabarovsk/24-1V/2022 (H5N1) virus were also demonstrated.

## 2. Materials and Methods

### 2.1. Strains of Viruses, Bacteria and Cell Cultures

Influenza viruses A/Astrakhan/3212/2020 (H5N8) (EPI_ISL_1038924), A/chicken/Astrakhan/321-05/2020 (H5N8) (EPI_ISL_1039234), A/chicken/Khabarovsk/24-1V/2022 (H5N1) (EPI_ISL_16618972), A/turkey/Stavropol/320-01/2020 (H5N8) (EPI_ISL_1114749), and A/chicken/Komi/24-4V/2023 (H5N1) (EPI_ISL_18525098) (FBRI SRC VB «Vector», Rospotrebnadzor) were used. Antigenic characterization of the viruses according to a hemagglutination inhibition (HI) assay is presented in the Section 3.

Plasmid DNA was obtained using *E. coli* Stbl3 strain (F′ proA+B+ lacIq ∆(lacZ)M15 zzf::Tn10 (TetR) ∆(ara-leu) 7697 araD139 fhuA ∆lacX74 galK16 galE15 e14-Φ80dlacZ∆M15 recA1 relA1 endA1 nupG rpsL (StrR) rph spoT1 ∆(mrr-hsdRMS-mcrBC)).

The microneutralization assay was performed using the MDCK-SIAT1 cell line, kindly provided by the WHO Collaborating Centre for Reference and Research on Influenza, The Francis Crick Institute. This cell line was used in all of the microneutralisation experiments.

The viruses produced in chick embryos were used in the challenge studies.

### 2.2. Development of DNA Vaccine pVAX-H5

The design, development, and production of the DNA vaccine pVAX-H5 is described in our previous work [31]. Briefly, for mice immunization, plasmid DNA was isolated using the EndoFree Plasmid Giga Kit (Qiagen, Hilden, Germany) and dissolved in saline. A quantitative and qualitative assessment of isolated plasmid DNA was carried out using a NanoDrop™ OneC spectrophotometer (Thermo Fisher Scientific, Waltham, MA, USA) at wavelengths of 260, 280, and 230 nm. Endotoxin detection was carried out using the LAL reagent Endosafe-PTS (Charles River Laboratories, Wilmington, MA, USA) according to the manufacturer’s instructions.

### 2.3. Experimental Design

The Guide for the Care and Use of Laboratory Animals was used for the animal experiments. The Animal Care and Use Committee (IACUC) at the State Research Center of Virology and Biotechnology “Vector” established the protocols for work with animals (BEC Protocol No. 1 of 03/21/2023).

Female BALB/c mice (16–18 g) were used for immunization. Immunization preparations were dissolved in 50 μL of physiological solution and were injected into the area of the left hind paw by jet injection using a Comfort-IN injector (Amcal, Perth, Australia) with individual nozzles. A detailed description of the immunization protocol using jet injection is described in our previous work [31].

The experiment consisted of two parts. The first part of the experiment was to assess the humoral response to the different doses of the experimental DNA vaccine. Seven groups (6 animals in each group) were involved in the study. Groups 1–5 were injected with 1, 10, 50, 100, and 200 μg of pVAX-H5 solution in saline, respectively. Group 6 was injected intramuscularly with 50 μg of recombinant hemagglutinin in complex with incomplete Freund’s adjuvant as a positive control; Group 7 included the intact animals. Immunization was performed twice with an interval of 21 days. Fourteen days after the second immunization, blood was taken from the retrobulbar sinus of the animal’s eye to analyze the humoral response. Animals were taken out of the experiment by the cervical dislocation method.

The second part of the experiment was to assess the protective properties of the experimental DNA vaccine. Four groups (10 animals in each group) were involved in the study. Groups 1–2 were injected with 100 μg of pVAX-H5 solution in saline; Groups 3–4 were injected with 100 μg of pVAX1 in saline. Immunization was performed twice, with an interval of 21 days. Fourteen days after the second immunization, blood was taken from the retro-orbital sinus.

### 2.4. Enzyme-Linked Immunosorbent Assay (ELISA)

ELISA was performed on the immunosorbent of recombinant hemagglutinin of influenza virus A/turkey/Stavropol/320-01/2020 (H5N8) (FBRI SRC VB «Vector») [32]. A detailed description of the ELISA protocol is described in our previous work [31].

### 2.5. HI and Microneutralization Assays

The HI assay was carried out as described in [32]. Standardized influenza virus was prepared by diluting each virus strain to a final HA titer of 4 HA units/25 µL. The animal sera were treated with receptor-destroying enzyme (RDE) and heat-inactivated prior to the assay. Then 2-fold dilutions of serum were mixed with 25 µl of standardized influenza virus (each well contained a final volume of 50 µL). The mixture was incubated at room temperature for 30 min. Then, 50 µL of 0.5% turkey erythrocyte suspension were added to each well. The assay was incubated at 4–8 °C until hemagglutination occurred in the control.

A microneutralization assay was performed as described in [32]. Briefly, each standardized virus contained 100 TCID_50_/100 µL. Two-fold dilutions of serum (200 µL) were mixed with 200 µL of standardized influenza virus. The suspensions were incubated for 1 h at 37 °C, 5% CO_2_. Non-immune mouse sera were used as the negative controls. Ferret reference sera (SRC VB Vector, Koltsovo, Russia) were used as the positive controls. Then, 200 µL of the suspension were added to the wells of culture plates with MDCK-SIAT1 cells. After 60 min, the inoculum was removed, and the cells were washed with a culture medium. The cells were cultured for 3 days in Opti-MEM I medium with 1 μg/mL TPCK-trypsin (Sigma-Aldrich, St. Louis, MO, USA) at 37 °C, 5% CO_2_. After that, the cells were stained with crystal violet solution, washed with water, and analyzed using an Agilent BioTek Cytation 5 multi-mode cell visualization reader (Thermo Fisher Scientific). All analyses were repeated in three replicates. The titer was taken as the serum dilution at which 50% of the cells survived. All negative controls had less than 5% surviving cells.

### 2.6. Challenge Studies

All challenge studies were carried out in accordance with the requirements of SanPiN3.3686-21 “Sanitary and Epidemiologic Requirements for the Prevention of Infectious Diseases” and Directive 2010/63/EC of the European Parliament and of the Council of the European Union of 22 September 2010, on the protection of animals used for research purposes.

Fourteen days after the second immunization, the mice were intranasally infected with the influenza A/Astrakhan/3212/2020 (H5N8) and A/chicken/Khabarovsk/24-1V/2022 (H5N1) viruses at a dose of 20 MLD50. 20 MLD50 = 6.5lg EID50 (embryonic infectious dose). Infection was carried out under anesthesia using a mixture of tiletamine and zolazepam and xylazine hydrochloride. After infection, the mice were monitored daily for 14 days, recording any manifestation of clinical symptoms of the disease, such as dishevelment, decreased body temperature, weight loss, neurological disorders, and cases of death. If severe conditions that could lead to the death of the animal occurred, such as loss of more than 20% of initial weight or lethargy, they were euthanized by cervical dislocation. The remaining mice were humanely killed in the same manner at the end of the experiment.

### 2.7. Statistical Analysis

GraphPad Prism 9.0 software (GraphPad Software, Inc., San Diego, CA, USA) was used for statistical analysis of the data obtained. Quantitative data are provided as the median with range and analyzed using nonparametric tests. The Mann–Whitney test was used in the case of two independent groups. A Kaplan–Meier multiplier estimator was used to construct the survival curve, and the Mantel–Cox test was used to compare the survival rates of the experimental and control groups.

## 3. Results

A DNA vaccine pVAX-H5 encoding a modified hemagglutinin of influenza virus A/turkey/Stavropol/320-01/2020 (H5N8) was previously developed, and its immunogenic properties were investigated [31]. A schematic comparison of the original amino acid sequence of the hemagglutinin of influenza virus A/turkey/Stavropol/320-01/2020 (H5N8) with the modified sequence in the DNA vaccine pVAX-H5 is shown in Figure 1.

In this study, the first step was to analyze the immunogenicity of different doses of DNA vaccine pVAX-H5. BALB/c mice were immunized with 1, 10, 50, 100, or 200 μg of pVAX-H5 twice at 3-week intervals using jet injection (Figure 2a). At 2 weeks after the second immunization, serum samples were obtained and analyzed by ELISA. All doses of vaccine pVAX-H5 induced the formation of specific antibodies to recombinant HA of the influenza virus A/turkey/Stavropol/320-01/2020 (H5N8) (Figure 2b). The median titers for different doses were: 1 μg—1:12150, 10 μg—1:36450, 50 μg—1:36450, 100 μg—1:72900, and 200 μg—1:36450. All vaccine doses also induced the formation of neutralizing antibodies against the corresponding A/turkey/Stavropol/320-01/2020 (H5N8) virus strain (Table 1). Based on these data, a dose of 100 μg of the pVAX-H5 DNA vaccine was selected for further studies.

The sera of animals immunized with 100 μg of pVAX-H5 were also analyzed in the microneutralization assay against three other strains of A/H5 clade 2.3.4.4b. All viruses were isolated from the sectional material of birds during outbreaks of highly pathogenic influenza in Russia in 2022–2024, including A/chicken/Komi/24-4V/2023 (H5N1) and A/chicken/Khabarovsk/24-1V/2022 (H5N1). The antigenic properties of the viruses are presented in Table 2.

As shown in the data presented in Table 3, the DNA vaccine pVAX-H5 based on the hemagglutinin of the A/turkey/Stavropol/320-01/2020 (H5N8) strain induced the production of antibodies that neutralized both the homologous strain A/Astrakhan/3212/2020 (H5N8) and the other strains used in the experiment. The sera reacted well with all viruses, with titers against A/chicken/Komi/24-4V/2023 (H5N1) ranging from 160 to 1280. The lowest neutralizing antibody titers were detected against influenza virus A/chicken/Khabarovsk/24-1V/2022 (H5N1) (40–80), which had low antigenic similarity with the vaccine virus A/Astrakhan/3212/2020 and the homologous virus A/turkey/Stavropol/320-01/2020. The HI titer of serum anti-A/Astrakhan/3212/2020 (H5N8) in HI with the A/chicken/Khabarovsk/24-1V/2022 virus is 20, whereas with a homologous virus, the titer is 160 (see Table 2). It should be noted that the HA in the DNA vaccine pVAX-H5 does not contain cytoplasmic and transmembrane domains (Figure 1).

A/Astrakhan/3212/2020 (H5N8) and A/chicken/Astrakhan/321-05/2020 (H5N8) viruses isolated during the H5N8 outbreak from humans and chickens, respectively, have identical HA protein sequences to the A/turkey/Stavropol/320-01/2020 (H5N8) virus, which was used for the studied DNA vaccine development. A/chicken/Khabarovsk/24-1V/2022 (H5N1) and A/chicken/Komi/24-4V/2023 (H5N1) have amino acid substitutions in HA. The complete amino acid sequences of the viral strains are given in the Appendix A.

In addition, the protective effect of pVAX-H5 (100 μg) was investigated upon infection with both A/Astrakhan/3212/2020 (H5N8) as a control and A/chicken/Khabarovsk/24-1V/2022 (H5N1). BALB/c mice were immunized with pVAX-H5 according to the scheme shown in Figure 3a. Before the second immunization and two weeks after the second immunization, animal serum samples were obtained for ELISA analysis. The results showed that the specific antibody titer after the first immunization was 1:4050. After the second immunization, the titer reached 1:109356, consistent with the previous experiment (Figure 3b). Mice were then infected with a lethal dose of virus A/Astrakhan/3212/2020 (H5N8) or virus A/chicken/Khabarovsk/24-1V/2022 (H5N1) (Figure 3a). When infected with a lethal dose of the heterologous A/chicken/Khabarovsk/24-1V/2022 (H5N1) virus, all vaccinated animals also survived, while all intact animals died or were euthanized by day 10 post-infection (Figure 3c). In a group of animals infected with A/Astrakhan/3212/2020 (H5N8), similar results were obtained, which is consistent with our previous results. These results indicate that the DNA vaccine induces cross-protective immunity in vivo.

## 4. Discussion

Global outbreaks of HPAI H5 viruses in wild birds and infection of other animals create the conditions for the virus to evolve in mammals, which could eventually lead to a virus strain capable of sustained human-to-human transmission, potentially resulting in a pandemic. This potential cross-species transmission requires continuous monitoring of avian influenza viruses and vaccine development [33,34,35,36]. A promising area for influenza vaccine development is nucleic acid-based vaccines: DNA and mRNA vaccines. These types of vaccines do not require the use of chicken embryos or cell cultures for production [37,38]. Determination of the nucleotide sequence of an actual virus strain is sufficient to initiate the development of a new vaccine [33].

In the development of nucleic acid-based influenza vaccines, it is important for an effective immune response to ensure that the antigen, in our case hemagglutinin, is presented in a prefusion conformation. It has been previously shown that stabilizing substitutions in pH-switch regions provide a closed conformation of the HA proteolysis site and prevent protein cleavage into subunits, which enhances the expression, quality, and stability of HA of influenza A and B viruses [39,40]. Therefore, in the development of the DNA vaccine, we modified HA by making stabilizing amino acid substitutions in the pH-switch region and added the trimerizing domain of phage T4. As a result, a DNA vaccine encoding the secreted HA A/turkey/Stavropol/320-01/2020 (H5N8) was developed, and its immunogenic properties were investigated against the homologous strain A/Astrakhan/3212/2020 (H5N8) 2.3.4.4b [31], which was recommended by the WHO as a vaccine strain [41]. However, antigenic drift and antigenic shifts result in changing influenza virus variants causing outbreaks, and therefore, it is important to determine whether existing and developing vaccines retain protective effects against new viral isolates [8]. For example, between 2019 and 2022, the major HPAI virus subtype causing global epizootics has changed from H5N8 to H5N1 [33,42].

This study aimed to investigate the dose-dependent effect and cross-reactivity of the DNA vaccine pVAX-H5. According to the comparative immunogenicity results, high levels of specific antibodies were observed starting from a dose of 50 μg (Figure 2). The highest ELISA titer of specific antibodies to recombinant HA was in the group receiving 100 μg. Notably, the higher dose of pVAX-H5 (200 µg) did not provide an increase in antibody titer compared to 100 µg (data are not significantly different, but there is a tendency to decrease in the case of 200 µg). The observed effect may be related to the fact that foreign DNA has adjuvant properties, activating innate immunity, which in turn, may negatively affect specific immunogenicity [43,44]. In the case of mRNA vaccines, dsRNA impurities have been shown to activate innate immunity, leading to a significant decrease in the immunogenicity of mRNA vaccines [45,46,47,48]. Therefore, dose escalation may be inappropriate or even harmful in the case of DNA vaccines administered by jet injection. For a more detailed understanding of the immune response to higher doses of pVAX-H5, doses in a wider range, e.g., up to 1 mg, and vaccination in other animal models should be investigated in the future.

The neutralizing activity against H5N1 strains was demonstrated for sera from animals immunized with 100 µg of pVAX-H5 using a microneutralization assay. Low neutralizing activity was observed against A/chicken/Khabarovsk/24-1V/2022 (H5N1), which correlated with a low HI titer of anti-A/Astrakhan/3212/2020 serum with the A/chicken/Khabarovsk/24-1V/2022 virus (1:20) (Table 3). Influenza A/H5 viruses, despite being within one clade 2.3.4.4b, may, in some cases, have significant antigenic differences according to previously reported HI studies. For example, a recent WHO report on clade 2.3.4.4b A(H5N1) influenza viruses isolated from cows and related human infections in the USA showed that few amino acid changes in the HA of candidate vaccine viruses (CVVs) may result in changes in their antigenic properties [49]. In this study, three different CVVs, including A/Astrakhan/3212/2020, A/American wigeon/South Carolina/22-000345-001/2021, and the strain recommended for cow vaccination, A/chicken/Ghana/AVL-763_21VIR7050-39/2021, were used in the antigenic analysis. Despite little difference in the amino acid sequences of HA (positions 104, 210, 511) of the CVVs, the titers in the HI test of the viruses and the post-infection ferret sera differed 4–8 fold from homologous titers for A/Astrakhan/3212/2020 and A/American wigeon/South Carolina/22-000345-001/2021 serum tested against A/chicken/Ghana/AVL-763_21VIR7050-39/2021 and some studied viruses. But, the antiserum A/chicken/Ghana/AVL-763_21VIR7050-39/2021 interacted very efficiently with all reference antigens (titers 2560) and studied viruses.

A similar broader cross-reactivity of an antiserum against studied viruses and lack of reciprocal reactivity in some cases was observed in our study. Antiserum to A/chicken/Khabarovsk/24-1V/2022 interacted with all antigens used in this study (titers 40-320). But the virus A/chicken/Khabarovsk/24-1V/2022 interacted with antiserum to vaccine strain A/Astrakhan/3212/2020 significantly worse (titer 20) than other antigens (titers 80-320).

It should be noted that the HI test shows antigenic differences based on the interaction of the virus with only antibodies that inhibit hemagglutination. In microneutralization, all antibodies that neutralize the virus are relevant. This study included both the HI test and microneutralization testing, thus providing a broader characterization of the antigenic properties of the viruses. It is also worth noting that there are no clear criteria for a protective titer for the microneutralization assay [50]. Thus, some researchers use a titer ≥80 as an endpoint for the efficacy of H5N1 avian influenza vaccines [51], while others consider a seroprotection threshold of 1:20 to be acceptable [52].

It was of particular interest to determine if pVAX-H5 would provide protection against infection with a lethal dose of A/chicken/Khabarovsk/24-1V/2022 (H5N1), since against this strain, the neutralizing titers were the lowest. It was shown that all animals immunized with 100 μg of pVAX-H5 survived, whereas all of the animals receiving the control plasmid pVAX1 died when infected with A (H5N1) strain (Figure 3c). It has also been previously shown that mRNA vaccines encoding HA clade 2.3.4.4b in vitro provide antibody formation that binds to different strains of the H5 subtype virus and protects mice from infection with clade 2.3.2.1a virus [53,54]. In this study, we did not analyze the specific T-cell response. However, in past work [31], we showed its formation in response to the administration of 100 μg of pVAX-H5 in similar settings. Therefore, we hypothesize that, in this case, in addition to neutralizing antibodies, T-cell immunity was involved in the formation of protective immunity. Recent studies by other groups have also shown that T-cell immunity induced by an mRNA-based vaccine contributes to protection against H5 virus infection [53,55,56].

Since mice as an animal model has limitations in immunological studies of human influenza, there is a need for further studies of the DNA vaccine in ferrets and possibly non-human monkeys. In addition, the protective properties of the DNA vaccine need to be studied against a broader spectrum of viruses in order to develop modifications that will increase the protection of the vaccine against heterologous viruses.

Thus, the data obtained indicate that the DNA vaccine encoding a modified hemagglutinin of influenza A(H5N8) virus is capable of protecting mice from infection with a lethal dose of the influenza A(H5N1) virus. The findings support the feasibility of DNA vaccine development for pandemic preparedness against HPAI viruses, which pose an increasing threat to human health.

## Figures and Tables

**Figure 1 viruses-17-00330-f001:**
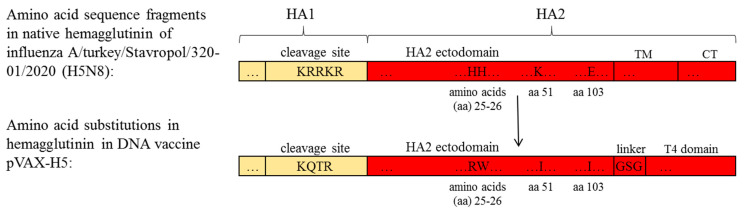
Schematic comparison of amino acid sequences of the original HA and the vaccine HA. TM—transmembrane domain, CT—cytoplasmic domain, T4 domain—trimerizing domain of phage T4.

**Figure 2 viruses-17-00330-f002:**
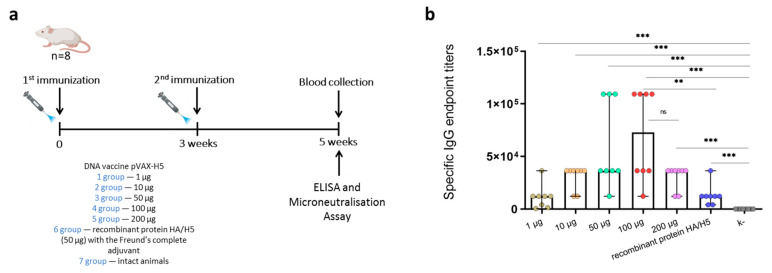
The humoral response to the different doses of the DNA vaccine pVAX-H5. (**a**) Experimental design. (**b**) Hemagglutinin-specific IgG antibody titer determined in immune sera by ELISA. The immunosorbent was recombinant hemagglutinin of influenza virus A (H5N8) [32]. Specific IgG endpoint titers are presented on the ordinate axis. Data are provided as median with range. The Mann–Whitney test was used for the statistical analysis of the data (**—*p* < 0.05; ***—*p* < 0.001; ns—not significant).

**Figure 3 viruses-17-00330-f003:**
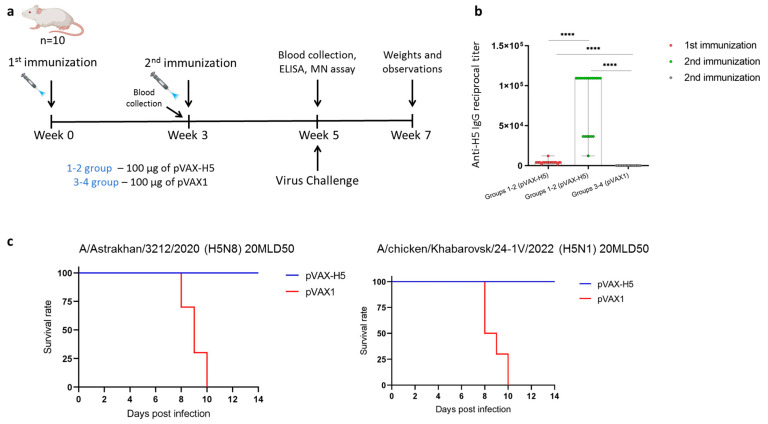
Evaluation of immunogenic and protective activity of DNA vaccine pVAX-H5. (**a**) Experimental design. (**b**) Hemagglutinin-specific IgG antibody titer determined in immune sera by ELISA. Data are presented as median with range. The Mann–Whitney test was used for statistical analysis (****—*p* < 0.0001). (**c**) Survival probability of immunized animals after infection with the influenza viruses of A/Astrakhan/3212/2020 (H5N8) and A/chicken/Khabarovsk/24-1V/2022 (H5N1) strains. Kaplan–Meier multiplier estimator was used to construct the survival curve, and the Mantel–Cox test was used to compare the survival rates of the experimental and control groups. pVAX-H5—a group of animals immunized with 100 μg of DNA vaccine pVAX-H5; pVAX1—group of animals immunized with 100 μg of plasmid pVAX1.

**Table 1 viruses-17-00330-t001:** Neutralizing activity of immune sera against A/turkey/Stavropol/320-01/2020 (H5N8) virus strain (reverse titer value).

Serum Code	1 μg	10 μg	50 μg	100 μg	200 μg	Recombinant Protein HA/H5	Intact Animals
**1**	320	2560	5120	5120	5120	540	<20
**2**	20	2560	1280	10240	5120	2560	<20
**3**	540	540	5120	10240	2560	2560	<20
**4**	1280	2560	5120	5120	2560	2560	<20
**5**	540	2560	2560	5120	5120	2560	<20
**6**	80	1280	5120	10240	10240	1280	<20
**7**	2560	2560	10240	2560	5120	2560	<20
**8**	2560	2560	5120	5120	5120	2560	<20

**Table 2 viruses-17-00330-t002:** Antigenic characteristics of A/H5 subtype influenza viruses by HI assay.

Viruses		Ferret Reference Serum				
		A/gyrfalcon/ Washington/ 41088-6/2014	A/chicken/Nghe An/27VTC/2018A/	A/gyrfalcon/ Washington/ 41088-6/2014	A/dalmatian pelican/Astrakhan/ 213-2V/2022	A/gyrfalcon/ Washington/ 41088-6/2014
**Reference antigens**	Subtype	Subclade				
		2.3.4.4c	2.3.4.4f	2.3.4.4b	2.3.4.4c	2.3.4.4b
**A/gyrfalcon/Washington/** **/41088-6/2014** **2.3.4.4.c**	H5N8	**320**	640	640	640	320
**A/chicken/NgheAn** **/27VTC/2018** **2.3.4.4.f**	H5N6	320	**640**	640	640	320
**A/Astrakhan/3212/2020** **2.3.4.4.b**	H5N8	80	80	160	80	80
**A/dalmatian pelican/Astrakhan/** **213-2V/2022** **2.3.4.4.b**	H5N1	160	320	320	**320**	320
**A/chicken/Khabarovsk/24-1V/2022** **2.3.4.4.b**	H5N1	20	80	20	80	**320**
**A/chicken/Komi/24-4V/2023** **2.3.4.4.b**	H5N1	320	<20	320	160	160
**A/chicken/Astrakhan/32105/2020** **2.3.4.4.b**	H5N8	80	80	160	80	80
**A/turkey/Stavropol/320-01/2020** **2.3.4.4.b**	H5N8	40	<20	80	80	40

**Table 3 viruses-17-00330-t003:** Virus-neutralizing activity of immune sera against influenza A virus strains H5Nx 2.3.4.4b (reverse titer value).

Serum Code	A/chicken/Astrakhan/321-05/2020 (H5N8)	A/chicken/Komi/24-4V/2023 (H5N1)	A/chicken/Khabarovsk/24-1V/2022 (H5N1)
1	1280	640	40
2	1280	160	40
3	640	320	80
4	2560	640	40
5	2560	640	40
6	1280	320	40
7	640	320	80
8	640	320	80
9	1280	320	80
10	1280	640	40
Control (+) *	2560	1280	640
Control (−) **	<20	<20	<20

* Ferret reference serum against each influenza virus was used as a positive control. ** a pool of sera from intact animals was used as a negative control.

## Data Availability

The data can be shared upon request.

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
