# Peer review of "Dose-Dependent Effect of DNA Vaccine pVAX-H5 Encoding a Modified Hemagglutinin of Influenza A (H5N8) and Its Cross-Reactivity Against A (H5N1) Influenza Viruses of Clade 2.3.4.4b"

_viruses, 2025, doi:10.3390/v17030330_

Round 1

Reviewer 1 Report

Comments and Suggestions for Authors

There is substantial interest in developing improved vaccines for influenza, including for HPAI H5 subtype viruses that pose a pandemic threat.  It has been known for a long time that DNA-based vaccines could provide broad protection for mice against diverse H5 viruses (e.g., in article PMID 9004608 from 1998, “Cross-protection among lethal H5N2 influenza viruses induced by DNA vaccine to the hemagglutinin”). DNA vaccines may potentially be manufactured faster than the currently licensed vaccines, including the recombinant HA-based ones, and there is certain interest in exploring new modifications of DNA vaccines and methods of their delivery.

In the manuscript “DNA vaccine encoding a modified hemagglutinin of influenza A/H5N8 virus protects mice from infection with a lethal dose of influenza A/H5N1 virus,” the authors used the previously published approach, the same DNA vaccine and its dose, to assess protection against a closely related H5 virus. The authors reported that the MN titers of the vaccinated mice were in a range 40-80 against the challenge virus, and this was sufficient to prevent the lethal outcome. No data are provided to demonstrate a difference in the morbidities (e.g., weight loss) when the challenge was done by the vaccine virus versus the other H5 virus.  

Although the data presented are interesting, the presentation of data should be improved, and terms and definitions used need clarification. Also, it would be appropriate/pertinent to provide additional data which could help to understand the correlation between HI, ELISA, and MN assays. It is also necessary to significantly shorten the manuscript as it contains an excessive review of literature on various aspects of vaccine design and delivery. The vaccine formulation used in this study was already published and there is no need to repeat the same points.  

Additional comments:

1.       Table 1. Antigenic characteristics of A/H5 subtype viruses by HI assay using ferret post-infection antisera. This table needs to be moved into Results section as the data seem to be used in defining antigenic relatedness among vaccine strains and other viruses in this study. It would be helpful to group these viruses according to their HA clade. Importantly, there is an apparent discrepancy between the outcomes of antigenic analysis when using ferret versus mouse sera for the same H5 viruses (Table 2). For example, A/chicken/Khabarovsk/2022 is antigenically similar to A/Astrakhan/3212/2020 virus based on HI assay (160 vs 80 titer), while these viruses seem to be antigenically different when analyzed using the sera from the vaccinated mice (640-2560 vs. 40-80). Therefore, it could be helpful to perform an additional antigenic analysis with ferret antisera using the microneutralization assay. Regardless of the outcome, the authors must provide a definition for the terms “homologous” and “heterologous” in the view of these discrepancies.

2.       Table 2. Neutralizing activity of sera from mice vaccinated with 100µg of DNA. In this table, it is necessary to provide clades of the viruses tested. The amino acid HA sequences of A/chicken/Astrakhan/321-05, A/turkey/Stavropol/320-01/2020 and the CVV strain A/Astrakhan/3212/2020 are identical. Therefore, it is unclear why A/chicken/Astrakhan/321-05/2020 is listed as “heterologous” virus. As in this study, only HA properties were investigated, there is no apparent reason for such distinction.

3.       Figure 1. Several doses of the DNA vaccine were used, and titers were determined using ELISA and microneutralization assays. However, MN results are not clear. It would be helpful to provide this analysis using a more conventional approach such as calculation of a geometric mean titer for each group.

It looks like, except for the group immunized with 1µg, the MN titers were similar. If the authors believe that ELISA provides more important data in the decision of what dose to use for immunization (100µg), then they need to provide ELISA results for the “heterologous” viruses (Table 2 contains only MN data for these viruses).    

a)       There is no need to repeat description of the groups at 1st and 2nd immunization as they are identical.

c)  The reviewer could not find sufficient information on microneutralization titers which were used to build this graph.

4.       Table 3. It would be beneficial to add information related to the HA sequence modifications in the DNA vaccine used in this study.

5.       Figure 2. a) no need to repeat the description of the groups at 1st and 2nd immunization as they were the same. c) the results from the challenge using A/Astrakhan/321/2020 serves as a control (this result was published previously); the only new data is protection after the challenge with A/chicken/Khabarovsk/24-1V/2022. It would be helpful to know TCID50 (the cell culture dose equivalent) for the 20 MLD50 used to challenge mice.

6.       In Abstract:

“…can adapt to mammals” – it would be prudent to state “are known to cause infections in mammals”

“With the constant evolution” – It would be more appropriate to state “due to ongoing evolution”

“Available and developing vaccine preparations” – please specify, vaccines for clade 2.3.4.4b or other clades as well?

“The highest titer” - need to clarify ELISA titers, as it is not the same for MN titers.

Strains names are provided, but no information is given for the respective HA clade/subclade.

“Prevented morbidity” – based on which parameters?

7.       Introduction:

A role of reassortment in generating viruses transmissible among humans and causing pandemics is not obvious.

The clade definition applies only to HA, not the gene constellation. Reassortment does not change HA clade.

8.       In the text: 50 MLD50 but in tables 20 MLD50.

9.       What was the original source of MDCK-SIAT1 cells? MDCK cells are also mentioned but not clear if and when this cell line was used.

10.   What are “mean median titers”?  

11.   “a slight rise” –  not clear how was it defined?

12.   “This strain also showed the lowest titers in HI assay (Table 1) and has the highest number of amino acid substitutions in HA relative to the vaccine strain (Table 2).”  – This description is not accurate. There is only a 2-fold difference between HI titers for CVV (Astrakhan) and the challenge virus (Khabarovsk) and this cannot be interpreted as the “lowest titers.”

Author Response

There is substantial interest in developing improved vaccines for influenza, including for HPAI H5 subtype viruses that pose a pandemic threat.  It has been known for a long time that DNA-based vaccines could provide broad protection for mice against diverse H5 viruses (e.g., in article PMID 9004608 from 1998, “Cross-protection among lethal H5N2 influenza viruses induced by DNA vaccine to the hemagglutinin”). DNA vaccines may potentially be manufactured faster than the currently licensed vaccines, including the recombinant HA-based ones, and there is certain interest in exploring new modifications of DNA vaccines and methods of their delivery.

In the manuscript “DNA vaccine encoding a modified hemagglutinin of influenza A/H5N8 virus protects mice from infection with a lethal dose of influenza A/H5N1 virus,” the authors used the previously published approach, the same DNA vaccine and its dose, to assess protection against a closely related H5 virus. The authors reported that the MN titers of the vaccinated mice were in a range 40-80 against the challenge virus, and this was sufficient to prevent the lethal outcome. No data are provided to demonstrate a difference in the morbidities (e.g., weight loss) when the challenge was done by the vaccine virus versus the other H5 virus.  

Although the data presented are interesting, the presentation of data should be improved, and terms and definitions used need clarification. Also, it would be appropriate/pertinent to provide additional data which could help to understand the correlation between HI, ELISA, and MN assays. It is also necessary to significantly shorten the manuscript as it contains an excessive review of literature on various aspects of vaccine design and delivery. The vaccine formulation used in this study was already published and there is no need to repeat the same points.  

We would like to thank the reviewer for careful and thorough reading of our manuscript and for the thoughtful comments and constructive suggestions, which help to improve its quality. Our response follows.

Corrections to the text are highlighted in green.

We shortened the introduction and removed some of the sentences in the discussion that are repetitive with our previous work

Additional comments:

  1. Table 1. Antigenic characteristics of A/H5 subtype viruses by HI assay using ferret post-infection antisera. This table needs to be moved into Results section as the data seem to be used in defining antigenic relatedness among vaccine strains and other viruses in this study. It would be helpful to group these viruses according to their HA clade. Importantly, there is an apparent discrepancy between the outcomes of antigenic analysis when using ferret versus mouse sera for the same H5 viruses (Table 2). For example, A/chicken/Khabarovsk/2022 is antigenically similar to A/Astrakhan/3212/2020 virus based on HI assay (160 vs 80 titer), while these viruses seem to be antigenically different when analyzed using the sera from the vaccinated mice (640-2560 vs. 40-80). Therefore, it could be helpful to perform an additional antigenic analysis with ferret antisera using the microneutralization assay. Regardless of the outcome, the authors must provide a definition for the terms “homologous” and “heterologous” in the view of these discrepancies.

We moved the table with the HI data to the ‘Results’ section and made a clarification regarding the clade of viruses used.

We rearranged the HI, and the refined data are shown in Table 2. In the tests to refine the HI results, we used erythrocytes depleted of the appropriate antigen.

We removed the data for A/chicken/Voronezh/193-1V/2024, as it was not possible to reproduce and interpret the data obtained.

We have made clarifications in the text when using the terms ‘homologous’ and ‘heterologous’ strains.

  1. Table 2. Neutralizing activity of sera from mice vaccinated with 100µg of DNA. In this table, it is necessary to provide clades of the viruses tested. The amino acid HA sequences of A/chicken/Astrakhan/321-05, A/turkey/Stavropol/320-01/2020 and the CVV strain A/Astrakhan/3212/2020 are identical. Therefore, it is unclear why A/chicken/Astrakhan/321-05/2020 is listed as “heterologous” virus. As in this study, only HA properties were investigated, there is no apparent reason for such distinction.

We agree with this comment and have made the necessary corrections to the text.

  1. Figure 1. Several doses of the DNA vaccine were used, and titers were determined using ELISA and microneutralization assays. However, MN results are not clear. It would be helpful to provide this analysis using a more conventional approach such as calculation of a geometric mean titer for each group.

It looks like, except for the group immunized with 1µg, the MN titers were similar. If the authors believe that ELISA provides more important data in the decision of what dose to use for immunization (100µg), then they need to provide ELISA results for the “heterologous” viruses (Table 2 contains only MN data for these viruses). 

Because the top point in MN was dilution 2560, we could not conclude on the difference in neutralizing activity between doses. We noted in the text that all groups showed neutralizing activity. We performed a refining experiment with the remaining sera in which we increased the dilution. The corrected data are summarized in Table 1.

Unfortunately, at this time we are unable to provide ELISA results for ‘heterologous’ viruses because we do not have a panel of appropriate recombinant hemagglutinins available. We tried to perform ELISA with virus lysates of these strains, but no representative data could be obtained.   

  1. There is no need to repeat description of the groups at 1stand 2nd immunization as they are identical.

The correction has been made. Please see Figure 2a.

  1. c)  The reviewer could not find sufficient information on microneutralization titers which were used to build this graph.

The correction has been made. The results of microneutralization were presented in the form of a table. Please see Table 1.

  1. Table 3. It would be beneficial to add information related to the HA sequence modifications in the DNA vaccine used in this study.

Table 4 provides information on amino acid substitutions present in other strains compared to the A/turkey/Stavropol/320-01/2020 (H5N8) strain used in the DNA vaccine design.

The DNA vaccine pVAX-H5 contains the hemagglutinin gene of strain A/turkey/Stavropol/320-01/2020 (H5N8) with the following modifications: the transmembrane and cytoplasmic domains were deleted and the fibritin trimerization domain of bacteriophage T4 was added; amino acid substitutions were made in the pH-switch regions of the HA2 subunit – H25R, H26W, K51I and E103I. An additional illustration is added where the differences between the original and vaccine HA are indicated. See Figure 1. The complete amino acid sequence of the viral strains is given in Supplementary 1.

  1. Figure 2. a) no need to repeat the description of the groups at 1stand 2nd immunization as they were the same. c) the results from the challenge using A/Astrakhan/321/2020 serves as a control (this result was published previously); the only new data is protection after the challenge with A/chicken/Khabarovsk/24-1V/2022. It would be helpful to know TCID50 (the cell culture dose equivalent) for the 20 MLD50 used to challenge mice.

The correction has been made. Please see Figure 2a.

The refinement was added to Materials and Methods: The virus was developed on chicken embryos. 20 MLD50 = 6.65lg EID50 (embryonic infectious dose).

  1. In Abstract:

“…can adapt to mammals” – it would be prudent to state “are known to cause infections in mammals”

The correction has been made.

“With the constant evolution” – It would be more appropriate to state “due to ongoing evolution”

The correction has been made.

“Available and developing vaccine preparations” – please specify, vaccines for clade 2.3.4.4b or other clades as well?

The correction has been made.

“The highest titer” - need to clarify ELISA titers, as it is not the same for MN titers.

The correction has been made.

Strains names are provided, but no information is given for the respective HA clade/subclade.

The correction has been made.

 “Prevented morbidity” – based on which parameters?

This conclusion was removed.

  1. Introduction:

A role of reassortment in generating viruses transmissible among humans and causing pandemics is not obvious. The clade definition applies only to HA, not the gene constellation. Reassortment does not change HA clade.

There may be other mechanisms for the emergence of pandemic influenza virus strains, but the role of reassortment has been proven, at least in analyses of pandemic influenza virus A/H1N1 pdm09. For example, Liu WJ, Wu Y, Bi Y, Shi W, Wang D, Shi Y, Gao GF. Emerging HxNy Influenza A Viruses. Cold Spring Harb Perspect Med. 2022 Feb 1;12(2):a038406. doi: 10.1101/cshperspect.a038406. PMID: 32928891; PMCID: PMC8805644.

  1. In the text: 50 MLD50 but in tables 20 MLD50.

We did an experiment where we investigated different doses of MLD50, but this paper only refers to 20 MLD50. We have corrected the inaccuracy. See Materials and Methods

  1. What was the original source of MDCK-SIAT1 cells? MDCK cells are also mentioned but not clear if and when this cell line was used.

The MDCK-SIAT1 cell line was kindly provided by the WHO Collaborating Centre for Reference and Research on Influenza, The Francis Crick Institute, London. This cell line was used in all microneutralisation experiments.

  1. 10.What are “mean median titers”? 

The correction has been made

  1. “a slight rise” –  not clear how was it defined?

We have rewritten this sentence.

  1. “This strain also showed the lowest titers in HI assay (Table 1) and has the highest number of amino acid substitutions in HA relative to the vaccine strain (Table 2).”  – This description is not accurate. There is only a 2-fold difference between HI titers for CVV (Astrakhan) and the challenge virus (Khabarovsk) and this cannot be interpreted as the “lowest titers.”

We have rewritten this fragment.

Reviewer 2 Report

Comments and Suggestions for Authors

In this study, Rudometov et al. examined the immunogenicity of a previously developed DNA vaccine that encodes a modified hemagglutinin from the influenza A/turkey/Stavropol/320-01/2020 (H5N8) virus. The vaccine was administered via jet injection at doses of 1, 10, 50, 100, and 200 μg. The results indicate that the DNA vaccine can protect mice from infection with a lethal dose of the influenza A/H5N1 virus. These findings substantiate the potential for developing this DNA vaccine as a strategy for pandemic preparedness against highly pathogenic avian influenza (HPAI) viruses, which increasingly threaten human health. However, certain aspects of the study require further clarification before publication.

1. Information regarding the viral strains, including the genome accession number, needs to be provided.

2. Despite the availability of reference materials, a brief description of the preparation of the DNA vaccine pVAX-H5 is necessary.

3. Line 144, Change CO2 to CO2.

4. Figure 1C indicates that the standard deviation for the 1 μg group is excessively high; please provide an explanation.

5. The limitations of this study need to be addressed in the discussion section.

Author Response

In this study, Rudometov et al. examined the immunogenicity of a previously developed DNA vaccine that encodes a modified hemagglutinin from the influenza A/turkey/Stavropol/320-01/2020 (H5N8) virus. The vaccine was administered via jet injection at doses of 1, 10, 50, 100, and 200 μg. The results indicate that the DNA vaccine can protect mice from infection with a lethal dose of the influenza A/H5N1 virus. These findings substantiate the potential for developing this DNA vaccine as a strategy for pandemic preparedness against highly pathogenic avian influenza (HPAI) viruses, which increasingly threaten human health. However, certain aspects of the study require further clarification before publication.

We would like to thank the reviewer for careful and thorough reading of our manuscript and for the thoughtful comments and constructive suggestions, which help to improve its quality. Our response follows.

Corrections to the text are highlighted in green.

  1. Information regarding the viral strains, including the genome accession number, needs to be provided.

We added the necessary information.

  1. Despite the availability of reference materials, a brief description of the preparation of the DNA vaccine pVAX-H5 is necessary.

We added the information to “Materials and Methods”:

Briefly, for mice immunization, plasmid DNA was isolated using the EndoFree Plasmid Giga Kit (Qiagen, Hilden, Germany) and dissolved in saline. Quantitative and qualitative assess-ment of isolated plasmid DNA was carried out using a NanoDrop™ OneC spectropho-tometer (Thermo Fisher Scientific, USA) at wavelengths of 260, 280, and 230 nm. Endotoxin detection was carried out using the LAL reagent Endosafe-PTS (Charles River Laboratories, Wilmington, MA, USA) according to the manufacturer’s instructions.

  1. Line 144, Change CO2 to CO2.

The correction has been made.

  1. Figure 1C indicates that the standard deviation for the 1 μg group is excessively high; please provide an explanation.

Because the top point in MN was dilution 2560, we could not conclude on the difference in neutralizing activity between doses. We noted in the text that all groups showed neutralizing activity. We performed a refining experiment with the remaining sera in which we increased the dilution. The corrected data are summarized in Table 1.

The correction has been made. The results of microneutralization were presented in the form of a table. Please see Table 1.

  1. The limitations of this study need to be addressed in the discussion section.

We have tried to note all the limitations of this study in the discussion section.

Round 2

Reviewer 1 Report

Comments and Suggestions for Authors

The manuscript was not sufficiently revised.

The use of the same vaccine in mice was already published and is not a new information.

Ther is insufficient characterization of the heterologous virus used for the challenge (A/chicken/Khabarovsk/...). For example, there are few amino acid differences in the HA of this virus compared to the vaccine, and yet there is a drastic difference in microneutralization titers. None of the HA mutations are in the key antigenic sites which raises a question. Based on the HA sequence alone, no one would be surprised to see a good protection from lethality. The HA mutations are not discussed in this lengthy manuscript. 

Comments on the Quality of English Language

Improvements are needed. 

Author Response

We would like to thank the reviewer for careful and thorough reading of our manuscript and for the thoughtful comments and constructive suggestions, which help to improve its quality. 

We have changed the title of the manuscript, made changes to the text, and checked the English language.

Changes are highlighted in green.

Reviewer 2 Report

Comments and Suggestions for Authors

The quality of the manuscript has significantly improved.

Author Response

We would like to thank the reviewer for his careful and thorough reading of our manuscript, as well as for his insightful comments and constructive suggestions, which contributed to improving its quality.